# Quercetin Regulates the Integrated Stress Response to Improve Memory

**DOI:** 10.3390/ijms20112761

**Published:** 2019-06-05

**Authors:** Toshiyuki Nakagawa, Kazunori Ohta

**Affiliations:** 1Department of Neurobiology, Gifu University Graduate School of Medicine, Gifu 501-1194, Japan; 2Department of Health and Nutritional Science, Nagoya University of Economics, Aichi 484-8504, Japan; kohta@nagoya-ku.ac.jp

**Keywords:** GADD34, ATF4, amyloid-β, neurogenesis, Alzheimer’s disease

## Abstract

The initiation of protein synthesis is suppressed under several stress conditions, inducing phosphorylation of the α-subunit of the eukaryotic initiation factor 2 (eIF2α), thereby inactivating the GTP-GDP recycling protein eIF2B. By contrast, the mammalian activating transcription factor 4 (ATF4, also known as cAMP response element binding protein 2 (CREB2)) is still translated under stress conditions. Four protein kinases (general control nonderepressible-2 (GCN2) kinase, double-stranded RNA-activated protein kinase (PKR), PKR-endoplasmic reticulum (ER)-related kinase (PERK), and heme-regulated inhibitor kinase (HRI)) phosphorylate eIF2α in the presence of stressors such as amino acid starvation, viral infection, ER stress, and heme deficiency. This signaling reaction is known as the integrated stress response (ISR). Here, we review ISR signaling in the brain in a mouse model of Alzheimer’s disease (AD). We propose that targeting ISR signaling with quercetin has therapeutic potential, because it suppresses amyloid-β (Aβ) production in vitro and prevents cognitive impairments in a mouse model of AD.

## 1. Introduction

Alzheimer’s disease (AD) is a neurodegenerative disorder with increasing prevalence worldwide, and characterized by the deposition of neurofibrillary tangles and amyloid in the brain. These changes may affect memory and other cognitive functions [1]. Several other biological processes have been suggested to be dysregulated in AD, such as cholesterol/sterol metabolism, inflammation, and endosomal vesicle recycling [1]. In addition, it has been recently found that ER stress is activated in a mouse model of AD and in AD patients [2,3,4,5,6]. Several reviews have indicated that α-subunit of the eukaryotic initiation factor 2 (eIF2α)-mediated translational control regulates synaptic plasticity [7], that eIF2α phosphorylation is a molecular link between AD and diabetes [6], and that the integrated stress response (ISR) mediates memory impairments in AD associated with apolipoprotein E ε4 (ApoE4) [8].

Quercetin is a natural dietary flavonoid found abundantly in fruits and vegetables. There is evidence suggesting that quercetin exhibits antioxidant and anti-inflammatory activities, as well as anti-tumor properties [9]. Quercetin can also improve several pathological conditions such as diabetes [10] and AD through regulation of anti-oxidative stress enzymes via the action of nuclear factor (erythroid-derived 2)-like 2 (Nrf2) and the antioxidant effect of paraoxonase 2 (PON2) expression [11,12]. Quercetin and its metabolites have been detected in the rat brain after oral administration of quercetin [13]. Recently, several studies suggest the effects of quercetin on memory and cognition improvement may be associated with ISR regulation [14,15,16]. In this review, we discuss ISR signaling in AD and the effects of quercetin on memory and adult neurogenesis.

## 2. ISR Elements and Signaling

In neurons, many ribosomes are attached to the endoplasmic reticulum (ER), forming the rough ER. Protein synthesis occurs in the rough ER as well as in free ribosomes, and is regulated by ternary complexes between the methionyl initiator tRNA (met-tRNAi), eIF2α, and guanosine triphosphate (GTP). Initiation of translation is required for the formation of the ternary complex, which binds to the 40S ribosomal subunit and the eukaryotic initiation factors to form the 43 pre-initiation complex. The 43 pre-initiation complex, which binds to the 5′ untranslated region (UTR) of mRNA, scans the mRNA downstream to identify the initiation codon, AUG, by the anticodon of met-tRNAi. AUG recognition activates the GTPase-activating protein (GAP) eIF5 to convert the GTP bound to the eIF2α (eIF2α-GTP) in the ternary complex to GDP bound to eIF2α (eIF2α-GDP). After the release of eIF2α-GDP, the 60S ribosomal subunit joins its complex to produce an 80S initiation complex and elongation of peptides is performed after recruitment of elongation factors. The exchange of eIF2α-GDP for eIF2α-GTP is catalyzed by eIF2B, a guanine nucleotide exchange factor (GEF) responsible for the recycling of eIF2α-GTP to form a ternary complex with the methionyl initiator tRNA. The ternary complex is regulated by eIF2B because eIF2B activity is inhibited by the phosphorylation of eIF2α at the residue serine 51 (Ser51) [17]. Since eIF2α is more abundant than eIF2B, part of the phosphorylation of eIF2α affects eIF2B activity, resulting in the reduction of general protein synthesis because the phosphorylation of eIF2α at residue Ser51 regulates the rate of translation initiation. Paradoxically, mammalian activating transcription factor 4 (ATF4) is specifically translated [18]. ATF4 is a transcription factor targeting several genes involved in amino acid import and metabolism, redox, and mitochondrial function [19]. The phosphorylation of eIF2α at residue Ser51 is reversed by protein phosphatase 1c (PP1c), which is regulated by two interacting proteins: The constitutive repressor of eIF2α phosphorylation (CReP) [20] and the growth arrest and DNA damage-inducible gene 34 (GADD34), which is strictly regulated by stress [21]. When the phosphorylation of eIF2α at residue Ser51 is removed, ATF4 expression levels quickly decrease. Therefore, ATF4 expression is regulated by the phosphorylation status of eIF2α at Ser51. Under several stress conditions, eIF2α is phosphorylated at residue Ser51 by four protein kinases: General control nonderepressible-2 (GCN2) kinase, double-stranded RNA-activated protein kinase (PKR), PKR-endoplasmic reticulum (ER)-related kinase (PERK), and heme-regulated inhibitor kinase (HRI). These kinases are activated mainly by amino acid starvation, viral infection, ER stress, and heme deficiency, respectively. Activation of the above-mentioned four kinases under such conditions is known as the ISR (Figure 1) [22].

EIF2α phosphorylation and ATF4 expression are reduced in the hippocampus of GCN2^-/-^ mice. In CA1 hippocampal slices from GCN2^−/−^ mice, late long-term potentiation (L-LTP), which is required for transcription and translation, is induced by stimulation of a single 100 Hz train, but not by four 100 Hz trains, which elicit L-LTP in hippocampal slices from wild type mice. GCN2^−/−^ mice showed an impairment in contextual fear conditioning, but an intact auditory fear conditioning. Interestingly, a single training session per day, but not three training sessions, enhanced spatial learning evaluated by the Morris water maze [23]. This study demonstrated that gene transcription in the hippocampus is enhanced in GCN2^−/−^ mice because cAMP response element binding protein 2 (CREB)-targeting gene expression is increased when ATF4 expression is downregulated. Therefore, Costa-Mattioli et al. proposed that the plasticity of the synapse and gene transcription induced by CREB are suppressed at basal conditions, during which ATF4 expression is high, while during learning, suppression of CREB is removed by downregulation of ATF4 expression. Consistently, decreased eIF2α phosphorylation and enhanced spatial learning and memory were observed in eIF2α heterozygous mice with a mutation of the phosphorylation site at residue Ser51 (eIF2α^+/S51A^). Moreover, inhibition of dephosphorylation of eIF2α by a small molecule, Sal003, impairs synaptic plasticity in wild type mice, but not in ATF4 knockout mice. Sal003 infusion into the bilateral hippocampus of wild type mice impairs contextual fear memory while eIF2α phosphorylation is increased in the hippocampus [24]. These studies indicated that the levels of eIF2α phosphorylation and ATF4 expression are important for synaptic plasticity and memory formation.

## 3. ATF4 in Memory and Synaptic Plasticity

*ATF4*, also known as *cAMP response element binding protein 2* (*CREB2*) [25], *CREB-2* [26], or *tax-responsive enhancer element B67* (*TAXREB67*) [27], was originally cloned by screening with a DNA probe containing ATF-binding sites [28] and was later identified as a tax-responsive enhancer element in the LTR of HTLV-1 binding protein [27]. ATF4 has a leucine zipper region for protein interaction and a stretch of basic amino acids for DNA binding at the C-terminal, and belongs to the ATF/CREB protein family [26]. *ATF4* mRNA is widely expressed in mammalian tissues including the brain. *ATF4* is controlled translationally by regulated re-initiation [29,30]. Mouse *ATF4* mRNA has two upstream open reading frames (uORFs), uORG1 and uORF2, in the 5′ noncoding region. uORF1 and uORF2 encode three and sixty amino acid residues, respectively, and uORF1 localizes upstream of uORF2 and the ATF4 coding region. Currently, a model of ATF4 translation proposes that in the presence of a high number of ternary complexes of met-tRNAi and eIF2α-GTP in non-stressed conditions, ribosomes scan and translate uORF1 and reinitiate translation of uORF2. After translation of uORF2, ribosomes dissociate from the *ATF4* mRNA, leading to a reduction in ATF4 coding region translation. On the other hand, in cases of reduced levels of the ternary complex during stressed conditions, re-initiation of uORF2 translation is suppressed by a delay in the reacquisition of the ternary complex after translation of uORF1. Therefore, the ribosome scans and initiates translation of the ATF4 coding region [29,30,31]. ATF4 is degraded through the E3 ubiquitin ligase SCF (Skp1/Cullin/F-box protein) containing the β-transducin-repeat-containing protein (β-TRCP) [32], indicating that ATF4 expression is regulated by translation and post-translation. ATF4 heterodimerizes with Nrf2 to regulate heme oxygenase-1 (HO-1) expression [33]. Phosphorylation of ATF4 by protein kinase A regulates the expression of several genes such as the osteoclast differentiation factor Rankl [34]. ATF4 is phosphorylated by RSK2, the growth factor-regulated kinase whose mutation causes Coffin–Lowry syndrome that is associated with mental retardation and skeletal abnormalities [35]. ATF4 is essential for lens fiber cell differentiation [36]. These studies indicate that ATF4 is important for differentiation of bone and the lens, amino acid metabolism, and resistance to oxidative stress [22]. ATF4 also plays roles in several physiological processes such as memory [25,37]; that is, ATF4 binds CREB to control its activity [25], and expression of a dominant negative CREB2 improves spatial learning, indicating that ATF4 works as a memory-suppressor gene [37]. The ATF4 protein has been reported to be present in the axons of the brains of patients with AD. ATF4 is synthesized in the axon of primary hippocampal rat neurons exposed to amyloid-β (Aβ)_1–42_, which induces eIF2α phosphorylation in the axons, causing neuronal cell death [38]. Overexpression of ATF4 in the nucleus accumbens of rats showed an anxiolytic-like response. However, depression-like behavior was also observed [39]. On the other hand, knockdown of ATF4 in the mouse hippocampus resulted in an impairment of spatial memory, decreased spine and puncta of the PSD95 and AMPA receptor GluR1, indicating that ATF4 plays a key role in synapse formation and memory [40]. These studies indicated that exploration of the regulation of ATF4 expression is important for the treatment of several diseases. Recently, three mechanisms have been proposed. First, feedback inhibition by GADD34 wherein eIF2α is dephosphorylated, leading to a suppression of ATF4 expression and a recovery of protein synthesis [21]. Second, ATF6-induced p58^IPK^ expression, which has been identified as an inhibitor of the interferon-induced PKR, suppresses PERK activity, leading to the suppression of eIF2α phosphorylation and ATF4 expression [41,42]. Lastly, ER stress-induced ATF4 expression is suppressed by pretreatment with low doses of lipopolysaccharide (LPS), which activates toll-like receptor 4 signaling, independently of the suppression of the phosphorylation of PERK or eIF2α [43], like an ISR inhibitor (ISRIB) [44].

## 4. Integrated Stress Response and Alzheimer’s Disease

Aβ deposits in the brain, also known as senile plaques, are a product of Aβ precursor protein (APP) cleavage by γ-secretase. γ-Secretase is a protein complex containing presenilin (PS), which is critical for its activity and when mutated causes familial AD [45]. Aβ may enhance tau phosphorylation, suggesting a connection between senile plaques and the neurofibrillary tangles [1]. Phosphorylated PERK is detected in the hippocampus and the temporal lobe of AD patients by immunohistochemistry [2]. Phosphorylation of eIF2α has also been detected in the brain of AD patients by immunohistochemistry [46] and western blot analysis [3]. The major genetic factor for sporadic AD is the presence of the ApoE4 allele. eIF2α phosphorylation is associated with cognitive impairments in ApoE4 knock-in mice [4], which is rescued by PKR inhibition, coinciding with a reduction in ATF4 expression levels [47]. The expression levels of eIF2α phosphorylation are also increased in a mouse model of frontotemporal dementia overexpressing the P301L tau mutation. In this case, neuronal loss is rescued by a PERK inhibitor, which acts by suppressing the expression levels of eIF2α phosphorylation and ATF4 [48]. Increased levels of eIF2α phosphorylation are observed in the hypothalamus following intracerebroventricular (ICV) injection of AD-associated Aβ oligomers (AβOs) in mice and macaques, inducing glucose intolerance. This AβO-induced expression of eIF2α phosphorylation is attenuated by a TNF-α neutralizing monoclonal antibody, infliximab. Additionally, AβO-induced peripheral glucose intolerance is prevented by ICV injection of tauroursodeoxycholic acid (TUDCA), a chemical chaperone used to alleviate ER stress [49]. The expression levels of eIF2α phosphorylation and ATF4 increased around Aβ deposits in the brain of a mouse model of AD [15]. ER stress links both obesity and diabetes [50] and ISR is activated in the hypothalamus [51]. APP23 mice express a human *APP*_751_ cDNA with a Swedish double mutation on a C57BL/6 genetic background [52]. APP23 mice have been previously crossed with obese and diabetic db/db (*Lepr^db/db^*) mice to generate a mouse model of AD with obesity (APP23/*Lepr^db/db^*). ATF4 expression was increased in the cerebral cortex of APP23/*Lepr^db/db^* mice [15]. Taken together, these studies indicate that the ISR is activated in the brain of mouse models of AD and in AD patients.

Aβ production changes dynamically during sleep [53]. Sleep deprivation increases the levels of eIF2α phosphorylation as a consequence of PERK activation in the mouse cerebral cortex [54], and chronic sleep restriction causes an increase in both the number and the size of the Aβ deposits in the brain of a mouse model of AD [53]. These results suggest that sleep deprivation-induced ISR activation enhances Aβ secretion. Treatment with an ER stress inducer, tunicamycin, enhanced Aβ production by increasing the expression of PS1 and ATF4 [55]. ATF4 binds to the amino acid response element (AARE) [56], which is localized in the human *PS1* gene [57] (Figure 1). Although abundance of PS1 fragments is regulated by cellular factors [58], the levels of PS1 expression were also upregulated in the ISR by the binding of ATF4 to the regulatory region in the *PS1* gene. In ISR signaling induced by leucine and lysine deprivation, secreted Aβ was elevated dependently of ATF4 [57]. Macroautophagy (referred to as autophagy hereafter) involves an important cellular mechanism for the degradation of proteins, lipids, and organelles, such as mitochondria and peroxisomes by engulfing into a double-membrane-bound structure, the autophagosome. Fusion of autophagosomes with lysosomes forms an autolysosome to hydrolyze its contents. Autophagosome formation is regulated by sixteen *autophagy-related* (*atg*) genes, characterized in yeast [59]. In these genes, *ATG5* is essential for the formation of autophagosomes and autophagosome-lysosome fusion [60]. Since autophagy plays a role in the recycling of amino acids, glucose, and lipids, autophagy is induced when cells are exposed to amino acid starvation [61]. ISR signaling, eIF2α phosphorylation and ATF4 expression, is activated in autophagy impaired cells, which are generated by the knockdown of *ATG5* (Atg5^KD^), and in cells treated by chloroquine [62], which inhibits autophagic flux [63] because GCN2 activity is required to adapt to amino acid deprivation [64]. In Atg5^KD^ cells, secreted Aβwas elevated by PS1 induction through increased eIF2α phosphorylation and ATF4 expression. Aβ production was decreased by the addition of plant polyphenolic compounds, such as resveratrol that prevents GCN2 activity, resulting in the downregulation of eIF2α phosphorylation and ATF4 expression [62]. This evidence indicates that ISR regulates Aβ production.

## 5. Quercetin on Memory in AD Models

Accumulation of evidence suggests a protective role for quercetin in cognitive decline and neurodegenerative disease [11,72,73]. Here, we summarize recently reported studies on the effects of quercetin using a variety of AD models (Table 1), which show quercetin can improve cognition and memory, and have beneficial effects on AD in different species [65,66,67,68,69,70,71]. Karimipour et al. suggest that quercetin improves learning and memory through CREB activation and induces neurogenesis as a compensatory mechanism for neuronal cell death in the brain of Aβ_1–42_ injected rats [65]. Wang et al. speculate that quercetin improves the cognitive impairments present in an APP/PS1dE9 mouse model of AD, alleviating Aβ-induced mitochondrial dysfunction via regulation of AMP-activated protein kinase (AMPK) activity [68]. Cardona-Gomez’s group suggests an anti-inflammatory effect of quercetin in the hippocampus by the reduction of Iba-1 and iNOS microglial immunoreactivity in the CA1 area of the hippocampus [66], and suggest effects of quercetin on neuropathological changes, cognitive impairments, and anxiety present in aged 3xTg-AD mice [67]. Moreno et al. demonstrated that oral administration of quercetin-loaded nanoparticles improves the cognition and memory impairments in senescence accelerated mouse-prone 8 (SAMP8) mice [69]. Kong et al. showed that quercetin rescues the impairments in the climbing ability of a *Drosophila* AD model by inhibiting cyclin B expression [71]. These studies and several reviews [11,72,74] demonstrated the biological roles of quercetin and its potential molecular mechanisms against memory impairments in AD models.

## 6. Unfolded Protein Response and GADD34

Secreted and transmembrane proteins are generated and modified in the lumen of the ER where those proteins fold and form proper structures. When proteins are unfolded in the lumen, signals via three mechanisms are activated to induce gene transcription encoding chaperones and proteins related to ER-associated degradation (ERAD) and to inhibit translational initiation [75] (Figure 1). These signaling pathways responsible for controlling protein-folding homeostasis in the ER are called the unfolded protein response (UPR) [76]. PERK is required for the phosphorylation of eIF2α and the suppression of translation during ER stress. Perk deficiency causes susceptibility to cell death possibly by increasing IRE1 phosphorylation and caspase-12 activation [77]. Wolcott–Rallison syndrome is characterized by diabetes and mental retardation caused by human PERK gene mutations [78]. Prefrontal cortex-specific PERK deficient mice show enhanced behavioral perseveration and impaired behavioral flexibility, coinciding with low levels of eIF2α phosphorylation and ATF4 expression. These reductions in eIF2α phosphorylation and ATF4 expression in the prefrontal cortex are observed in patients with schizophrenia [79]. Activating transcription factor 6 (ATF6) is a type II transmembrane glycoprotein that is cleaved to produce a 50-kDa protein (p50ATF6) under ER stress conditions. The p50ATF6 translocates into the nucleus and binds to the ER stress-responsive element (ERSE) [80] to induce the translation of ER stress-related genes such as the *X-box binding protein 1* (*XBP1*) mRNA for UPR [81]. IRE1 is the most conserved transmembrane protein of the UPR, localizing at the ER. IRE1 has a stress sensing ER luminal domain, a cytosolic kinase, and a sequence specific endoribonuclease domain. *XBP1* mRNA is spliced by IRE1 after activation via oligomerization. During ER stress conditions, *GADD34* mRNA is induced to transcribe and translate the GADD34 protein from its ORF through leaky scanning of uORF1 and uORF2 by the ribosomes [82]. GADD34 was originally identified as an ionizing radiation-inducible transcript in Chinese hamster ovary (CHO) cells, and has been identified as the homolog of the mouse MyD116 [83]. GADD34 has three domains: An ER-targeting domain, four central PEST domains, and a C-terminal PP1c-binding domain. The GADD34:PP1c holoenzyme recruits phosphorylated eIF2α for dephosphorylation, mediated by PEST domains [84]. Promotion of dephosphorylation of eIF2α by the expression of GADD34, decreasing the levels of eIF2α phosphorylation and ATF4 expression, improves synaptic function and prevents neuronal degeneration in prion-infected mice [85]. Therefore, GADD34-mediated dephosphorylation of eIF2α inhibits the UPR, leading to the recovery from translational suppression in a negative feedback manner [21]. Quercetin has been found to interact with the ligand-binding pocket at the dimer interface of the kinase extension nuclease domain of IRE1 for activation of its ribonuclease [86]. Consistently, *XBP1* mRNA splicing is enhanced in quercetin-treated cells, coinciding with an increase in *GADD34* mRNA transcription [55]. Quercetin has been shown to suppress eIF2α phosphorylation, ATF4 expression, and Aβ production. These findings were not observed in GADD34 knockdown cells. These studies indicate that quercetin activates IRE1 and regulates GADD34 expression to control ISR, although more experiments are required for the identification of the molecular mechanisms associated with GADD34 induction through IRE1.

## 7. Quercetin Improves Memory Impairments in Mouse Models of Alzheimer’s Disease by Adjusting the Integrated Stress Response

According to the amyloid cascade hypothesis, Aβ formation is a critical step in the progression of AD [87]. Understanding the role of the ISR in the brain can contribute to the elucidation of the fundamental link between AD pathogenesis and the cause of morbidity. Axonally translated ATF4 protein leads to cell death in response to Aβ [38]. eIF2α phosphorylation plays a crucial role in long-lasting synaptic plasticity and memory consolidation through the regulation of gene expression and the translational control of *ATF4* mRNA [23,24,88]. High ATF4 expression in the brain of APP23/*Lepr^db/db^* mice showed an impairment of short-term episodic-like memory [16]. Increased levels of the n-terminal fragment of presenilin-1 (PS1) and ATF4 expression were observed in the brain of mice fed a leucine- and lysine-deficient diet (LLD). These mice showed impairments in working memory, as assessed by a Y-maze experiment (Figure 2). Deficiency of essential amino acids such as leucine in the diet accumulates uncharged tRNA, inducing GCN2 activation by binding with the uncharged tRNA to phosphorylate eIF2α in the mouse brain. This GCN2/P-eIF2α signaling is essential for denial of an essential amino acid-deficient diet within 20 min of the behavioral response to survive [89]. In contrast, mice fed LLD for a long time resulted in an induction of ISR signaling in the brain, leading to an increased ATF4 expression and an impairment of working memory. Moreover, mice bilaterally infused in the hippocampus with a derivative of salubrinal, which blocks eIF2α dephosphorylation, showed impaired contextual memory [24]. In contrast, an ISR inhibitor (ISRIB), which activates the guanine nucleotide exchange factor (GEF) of eIF2B [90], enhanced spatial and fear-associated learning [44]. These findings strongly suggest that the eIF2α phosphorylation-ATF4 signaling cascade is involved in memory impairment and that small molecules that modify the ISR activity are important to improve memory function.

Oral administration of 0.5% quercetin for 5 weeks reduced the levels of ATF4 expression in the hippocampus, amygdala, and cerebral cortex of the brain of APP23/*Lepr^db/db^* mice [16]. This reduction in ATF4 expression was observed in the brain of APP23 mice fed quercetin for more than one year [15]. In these conditions, fear-associated learning (contextual and auditory fear conditioning) in wild-type mice (aged 1 year ± 6 months) was improved after oral administration of 0.5% quercetin for 20 weeks [15]. The percentage of auditory fear conditioning in APP23 mice that were fed quercetin in the long-term (from 4 to 60 weeks old), was examined every two months. Although there was no difference in each auditory fear conditioning test between the APP23 mice on a basal diet and those on a basal diet supplemented with quercetin, the deterioration in memory was delayed in aged but non-diabetic AD mice fed with quercetin (aged 6–12 months). Contextual and auditory fear memories were enhanced in aged wild-type mice fed with quercetin [15]. In mice fed with quercetin, GADD34 expression was increased, coinciding with a reduction in ATF4 expression [15]. In high-cholesterol-fed old mice, quercetin improves cognitive impairment, suppressing eIF2α phosphorylation [14]. Furthermore, memory was consistently enhanced in *eIF2α* heterozygous mice (eIF2α^+/S51A^) [24]. These findings demonstrate that quercetin influences ISR signaling by inducing the expression of GADD34 in the brain to improve memory function.

## 8. Quercetin in Adult Neurogenesis

The adult mammalian brain contains neural stem cells (NSCs) located in the subventricular zone (SVZ), hippocampal subgranular zone (SGZ) [91], and in the hypothalamic subependymal niche [92,93]. NSCs generate neurons and glial cells [94], which may be important for the homeostasis of tissues. Interestingly, neurogenesis also plays critical roles in learning and memory [91]. For example, in the hippocampal dentate gyrus, neurogenesis is important for spatial learning [95] and pattern separation, the process whereby the brain discriminates between two similar objects [96,97]. New neurons reach the olfactory bulb from the subventricular zone via the rostral migratory stream [98]. This instance of neurogenesis plays a role in olfactory learning [99]. Hypothalamic neurogenesis, which is important in the regulation of food intake [100], decreases with aging, and hypothalamic NSCs control aging speed partly by releasing exosomal microRNAs [101]. It is also suggested that adult hippocampal neurogenesis may be dysregulated in a mouse model of AD and in AD patients [85,102] and decreases during aging [103]. Using induced pluripotent stem cells (iPSCs), it has been shown that neural differentiation is accelerated in iPSCs-derived neural cells from sporadic AD patients and progenitor cell renewal is reduced through a decrease in the levels of the repressor element 1-silencing transcriptional factor (REST) [104]. These studies indicate that regulation of neurogenesis may be critical for the improvement of learning.

Neurogenesis is regulated by several factors, including growth factors [99]. Voluntary running increases SGZ neurogenesis, which is enhanced by neurotrophic factors such as brain-derived neurotrophic factor (BDNF) [95]. Wrann et al. showed that BDNF is induced in the hippocampus by exercise, which is mediated by irisin [105]. Irisin is a cleaved product of fibronectin type III domain-containing protein 5 (FNDC5), which is induced by exercise in muscles [106] and the hippocampus [107]. Irisin improves obesity, glucose homeostasis [106], and memory impairments in an AD mouse model [107]. Irisin also enhances the CREB pathway in human cortical slices and prevents amyloid-β oligomer (AβO)-induced eIF2α phosphorylation and ATF4 expression in cultured primary rat hippocampal neurons [107]. Recently, it has been shown that, similar to the effects of exercise, increasing adult neurogenesis with BDNF induction improves memory in an AD mouse model. By contrast, ablation of adult hippocampal neurogenesis leads to an increase in cognitive impairments in older, but not younger, 5xFAD mice. These results suggest that the regulation of adult neurogenesis and BDNF expression may be valuable for improving memory function and modulating the progression of AD [108]. In a mouse model of obesity, the increased numbers of senescent glial cells caused by fat deposits in obesity leads to impairment of neurogenesis in the lateral ventricle (LV), and mice also show higher levels of anxiety. In this case, clearing senescent cells using a senolytic drug cocktail containing dasatinib and quercetin rescues anxiety and increases neurogenesis [109]. Administration of quercetin-3-*O*-glucuronide, which is a major quercetin metabolite [110], increases adult hippocampal neurogenesis in mice [111]. Administration of 14~16 mg of quercetin to rat daily for one month can promote the proliferation and differentiation of NSCs, increasing the expression of *BDNF*, *NGF*, *CREB*, and *Zif268* mRNAs [65]. Quercetin prevents the reduction of PGC-1α, FNDC5, and BDNF expression in the hippocampus of rats exposed to hypobaric hypoxia [112], and increases the number of doublecortin (DCX)-expressing cells in the adult rat dentate gyrus of the hippocampus, inducing BDNF mRNA expression [65]. Neurogenesis in the hippocampal dentate gyrus is important for discrimination of two similar things such as pattern separation evaluated by fear conditioning using two chambers [97]. Mice fed quercetin showed a higher percentage of freezing compared with mice fed a control diet, although mice fed quercetin as well as the control diet could not discriminate between the two chambers [113]. These studies suggest that quercetin may have a positive effect in neurogenesis and BDNF expression (Figure 3), and may lead to the slowing of the progression of early-stage AD. 

## 9. Conclusions and Perspective

The ISR mediates cognitive impairments in mouse models of AD. The signaling molecules of the ISR were increased in post-mortem AD brains and in other neurological diseases such as traumatic brain injury, amyotrophic lateral sclerosis, Huntington’s disease, and Parkinson’s disease [114]. Inhibition of ISR signaling by quercetin rescues memory deficits in mouse models of AD and ISRIB prevents cognitive deficits in traumatic brain injury [115] and neurodegeneration in prion-disease mice [116]. However, identification of the target molecules involved in the ISR signaling in these neuronal diseases for treatment remains elusive. Future experiments should explore the main molecules for potential therapeutic interventions such as eIF2α kinases, eIF2B, and phosphatases (PP1c, CReP, and GADD34).

## Figures and Tables

**Figure 1 ijms-20-02761-f001:**
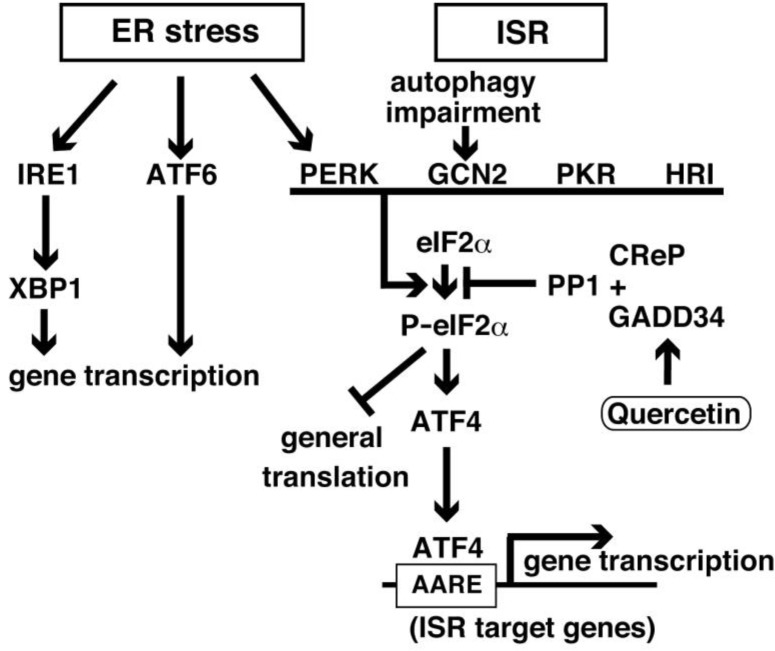
Signaling cascades in integrated stress response (ISR) and endoplasmic reticulum (ER) stress. IRE1: Inositol-requiring enzyme 1; XBP1: X-box binding protein 1; ATF6: Activating transcription factor 6; and PP1: Protein phosphatase 1. Arrow: activation; Flat line: inhibition.

**Figure 2 ijms-20-02761-f002:**
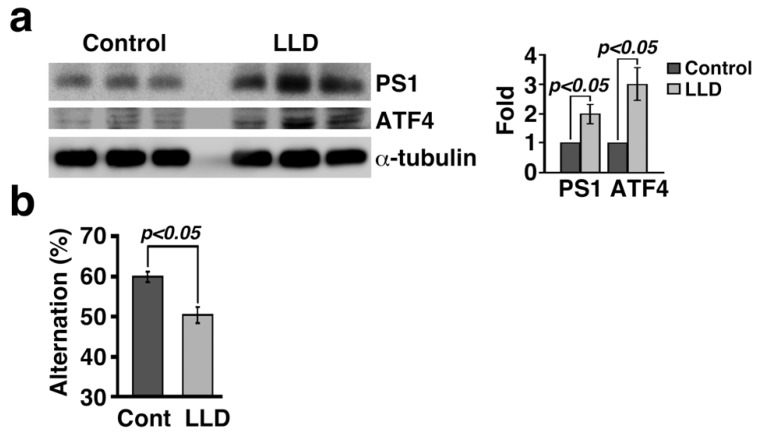
Working memory is impaired in mice fed with amino acid imbalanced food. (**a**) PS1 and ATF4 expression are significantly increased in the brain of mice fed with leucine- and lysine-deficient food (LLD) 30 minutes per week, for ten times. (**b**) Percentage of alternation measured in the Y-maze test decreased in mice fed an LLD. The protocol of animal study was approved by the Gifu University Graduate School of Medicine Animal Care and Use Committee of 20-131 (15/1/2009).

**Figure 3 ijms-20-02761-f003:**
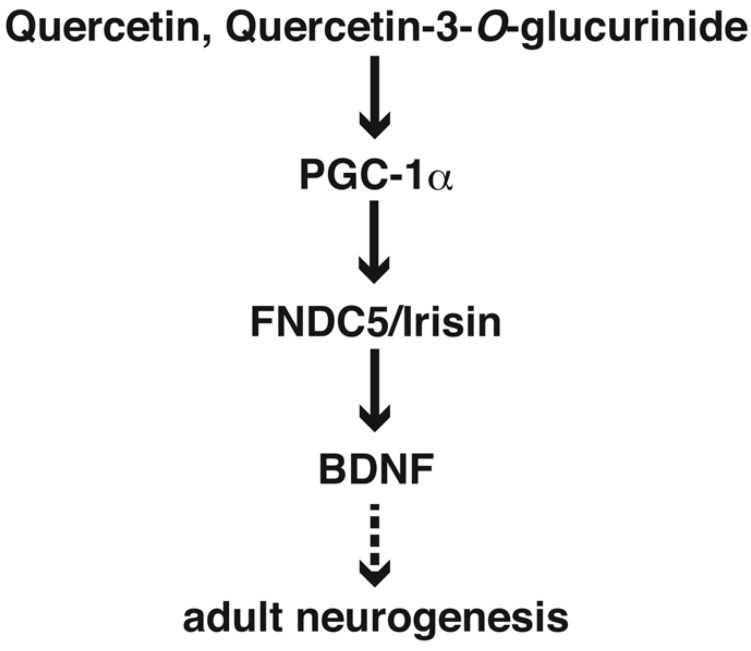
Quercetin and its metabolite may induce adult neurogenesis through the PGC-1α-FNDC5/Irisin-BDNF cascade.

**Table 1 ijms-20-02761-t001:** Alzheimer’s disease (AD) model and effects of quercetin on memory.

AD Model	Dose and Duration of Quercetin	Effects	References
3 µg/µL of Aβ1–42 injection into bilateral intracerebroventricular zones of rats	40 mg/kg/day orally, one month	improvement of spatial learning and memory. increase of the number of doublecortin (DCX)-expressing cells in the dentate gyrus. increase of BDNF expression.	[65]
3xTg-AD mice	25 mg/kg intraperitoneal injection, every 48 h for three months	reduction of Iba-1 and iNOS microglial immunoreactivity in the CA1 area of the hippocampus. decreased fluorescence intensity of Aβ.	[66]
3xTg-AD mice	25 mg/kg intraperitoneal injection, every 48 h for three months	decrease of Aβ, tauopathy, astrogliosis, and microgliosis in the hippocampus and the amygdala. improvement of spatial learning, memory, and anxiety.	[67]
APPSWE/PS1dE9 mice	40 mg/kg/day orally, 16 weeks	improvement of mitochondria dysfunction. increase AMPK activity.	[68]
Senescence Accelerated Mouse-Prone 8 (SAMP8) mice	25 mg/kg quercetin-loaded nanoparticles (NPQ) orally, every two day, two months	improvement of the cognition and memory impaiments by NPQ. decreased expression of the hippocampal GFAP expression.	[69]
pentylenetrazole (PTZ)-induced cognitive impairment of zebrafish	10 mg/kg solid lipid nanoparticle of quercetin, single intraperitoneal injection	inhibition of PTZ-induced cognitive impairment and acetylcholinesterase activity.	[70]
human Aß expressing Drosophila	0.44 g/L in standard sugar-yeast medium, dietary supplementation of quercetin, 10 days	inhibition of Cyclin B expression. extended lifespan.	[71]

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
