# Peer review of "Quercetin Regulates the Integrated Stress Response to Improve Memory"

_ijms, 2019, doi:10.3390/ijms20112761_

Reviewer 1 Report

Comments and Suggestions for Authors

The paper entitled “Quercetin regulates the integrated stress response to improve memory” where the authors proposed that targeting ISR signaling with quercetin has therapeutic potential, because it suppresses amyloid-ß (Aß) production in vitro and prevents cognitive impairment in a mouse model of AD. It is an very interesting paper, well prepared, covering a lot of relevant information.

I have no specific scientific concern but it is mandatory to correct the manuscript in some points:

#line 87:

Is ‘Quercetin improves cognition and memory disturbed by Aß or pentylenetetrazole in AD models, such as rodents[18-21], zebrafish[22], or Drosophila[23]. Karimipour et al., suggest that quercetin improves learning and memory through CREB activation and induces neurogenesis as a compensatory mechanism for neuronal cell death in the brain of amyloid-ß (Aß)1-42 injected rats[19].’

Should be ‘Quercetin improves cognition and memory disturbed by Aß or pentylenetetrazole in AD models, such as rodents [18-21], zebrafish [22], or Drosophila [23]. Karimipour et al., suggest that quercetin improves learning and memory through CREB activation and induces neurogenesis as a compensatory mechanism for neuronal cell death in the brain of amyloid-ß (Aß)1-42 injected rats [19].’

I suggest that authors should insert a space between the reference and the sentence throughout the article, for example, is “…Drosophila[23].”, should be “…Drosophila [23].”.

#Table 1:

In the column references: reference numbers should be in parentheses.

#References:

Please read the instructions for the authors again because the literature is poorly prepared:

 „References should be described as follows, depending on the type of work:

  Journal Articles:
1. Author 1, A.B.; Author 2, C.D. Title of the article. Abbreviated Journal Name YearVolume, page range. Available online: URL (accessed on Day Month Year).

  Books and Book Chapters:
2. Author 1, A.; Author 2, B. Book Title, 3rd ed.; Publisher: Publisher Location, Country, Year; pp. 154–196.
3. Author 1, A.; Author 2, B. Title of the chapter. In Book Title, 2nd ed.; Editor 1, A., Editor 2, B., Eds.; Publisher: Publisher Location, Country, Year; Volume 3, pp. 154–196.

  Unpublished work, submitted work, personal communication:
4. Author 1, A.B.; Author 2, C. Title of Unpublished Work. status (unpublished; manuscript in preparation).
5. Author 1, A.B.; Author 2, C. Title of Unpublished Work. Abbreviated Journal Name stage of publication (under review; accepted; in press).
6. Author 1, A.B. (University, City, State, Country); Author 2, C. (Institute, City, State, Country). Personal communication, Year.

  Conference Proceedings:
7. Author 1, A.B.; Author 2, C.D.; Author 3, E.F. Title of Presentation. In Title of the Collected Work (if available), Proceedings of the Name of the Conference, Location of Conference, Country, Date of Conference; Editor 1, Editor 2, Eds. (if available); Publisher: City, Country, Year (if available); Abstract Number (optional), Pagination (optional).

  Thesis:
8. Author 1, A.B. Title of Thesis. Level of Thesis, Degree-Granting University, Location of University, Date of Completion.

  Websites:
9. Title of Site. Available online: URL (accessed on Day Month Year).
Unlike published works, websites may change over time or disappear, so we encourage you create an archive of the cited website using a service such as WebCite. Archived websites should be cited using the link provided as follows:
10. Title of Site. URL (archived on Day Month Year). “

Author Response

Point 1: I suggest that authors should insert a space between the reference and the sentence throughout the article, for example, is “…Drosophila[23].”, should be “…Drosophila [23].”.

Response 1: We are sorry for missing the space between the reference and the sentence. We inserted spaces between them in the revised text.

Point 2: #Table 1: In the column references: reference numbers should be in parentheses.

Response 2: We are sorry for the errors in the table. We amended them in the revised table (p 5). Please refer to Reviewer #2 Point 4.

Point 3: #References: Please read the instructions for the authors again because the literature is poorly prepared:

Response 3: We are sorry for the errors in the references. We amended them in the revised references.

Reviewer 2 Report

In this review by Toshiyuki Nakagawa et al., the authors summarized literatures on the role of integrated stress response (ISR) signaling in a mouse model of Alzheimer’s disease, and the effects of quercetin in memory and adult neurogenesis. The topic on evidence and mechanisms linking ISR to memory impairment in AD is interesting and therapeutically promising. Several review articles has summarized the roles for ISR elements in cognitive decline in either AD or non-AD contexts (PMID: 25032491, PMID: 25857551). A recent review also summarized the pharmacological effects of flavonoids, including quercetin, in animal models for AD and Parkinson’s disease model (PMID: 29861833). This review has Table 1. summarizing literatures about the effects of quercetin on memory in AD mouse model. The authors want to propose that the effects of quercetin on memory improvement is through IRS inhibition. There are several concerns on the clarity and readability of this review. The contents and layout of the review need to be re-organized before accept for publication.

1.       Introduction. This is the first part of review, and is beginning with the cellular events/activation of each IRS elements. Although Figure 1 is there to help illustrate signaling cascades of ISR and ER stress, the description in the text is still difficult for reader to follow. I suggest that the first paragraph of introduction should begin with AD and the potential of targeting on IRS with quercetin to improve memory. Then followed by a separate subtitle of 1. IRS elements and signaling.

2.       Suggested subtitle 2: ATF4 in memory and synaptic plasticity. This paragraph should focus on introducing the expression and regulation of ATF4 in brain and neurons under AD or non-AD context, and how it is functionally correlated to memory and synaptic plasticity. 

3.       3. ISR and AD, 4. ISR and A-beta production can be combined under one subtitle. Authors should refer to the previous reviews (PMID: 25032491, PMID: 25857551, PMID: 26818496), and focus on more recent papers on this topic since the above mention reviews published in 2016.

4.       A summary of quercetin on memory in AD mouse model, and table 1 can be moved to this separate subtitle. Authors should make sure all the literature on this topic is included in the table. Eg. PMID: 25666032.

5.       The molecular mechanism underlying the protective role of quercetin in AD is proposed via the antioxidant pathways by other reviews (PMID: 30914316, PMID: 26904161), which should be mentioned and discussed in this review. This current review tries to link a novel mechanism pathway of quercetin via IRS inhibition, based on their previous finding that quercetin suppress ATF4 expression in an AD and diabetic mouse model (Ref. 52). Does quercetin suppress ATF4 in aged but non-diabetic AD mice? In the same paragraph, other evidence that quercetin may regulate memory of AD or non-AD model through the IRS pathway should be stated with correlated reference.  

6.       It is best to include a paragraph of “conclusion and Perspective” in the end of the review article.

7. The abstract stated that IRS signaling in human AD patients is also reviewed, but there is no clearly part about human study reviewed in the text. 

Author Response

Point 1: Introduction. This is the first part of review, and is beginning with the cellular events/activation of each IRS elements.---. I suggest that the first paragraph of introduction should begin with AD and the potential of targeting on IRS with quercetin to improve memory. Then followed by a separate subtitle of 1. IRS elements and signaling.

Response 1: We agree with the Reviewer’s comment. We mentioned in the first paragraph of the revised text (p 1) that “Alzheimer’s disease (AD) is a neurodegenerative disorder----” and followed it by a separate subtitle “2. IRS elements and signaling”.

Point 2: Suggested subtitle 2: ATF4 in memory and synaptic plasticity. This paragraph should focus on introducing the expression and regulation of ATF4 in brain and neurons under AD or non-AD context, and how it is functionally correlated to memory and synaptic plasticity. 

Response 2: We agree with the Reviewer’s comment and mentioned in the revised text (p 1) that “3. ATF4 in memory and synaptic plasticity -----”.

Point 3-1: 3. ISR and AD, 4. ISR and A-beta production can be combined under one subtitle.

Response 3-1: We agree with the Reviewer’s comment. We combined subtitles 3 and 4 and mentioned in the revised text (p 4) that “4. Integrated stress response and Alzheimer’s disease -----”.

Point 3-2: Authors should refer to the previous reviews (PMID: 25032491, PMID: 25857551, PMID: 26818496), and focus on more recent papers on this topic since the above mention reviews published in 2016.

Response 3-2: We are sorry for missing these citations. We cited papers (PMID: 25032491, PMID: 25857551, PMID: 26818496) in the revised text (p 1) and mentioned that “Several reviews have indicated that the α-subunit of the eukaryotic initiation factor 2 (eIF2α)-mediated translational control regulates synaptic plasticity (PMID: 25032491), eIF2α phosphorylation is a molecular link between AD and diabetes (PMID: 25857551), and the integrated stress response (ISR) mediates memory impairment in AD associated with ApoE4 (PMID: 26818496)”.

Point 4: A summary of quercetin on memory in AD mouse model, and table 1 can be moved to this separate subtitle. Authors should make sure all the literature on this topic is included in the table. Eg. PMID: 25666032.

Response 4: We agree with the Reviewer’s comment. We moved Table 1 and the text to separate the subtitle in the revised text (p 5) as follows: “5. Quercetin on memory in AD models ---”. We are sorry for missing this citation. We mentioned in the revised text (p 5) that “The role in memory and the molecular mechanisms of quercetin have been reviewed (Zaplatic, et al., Life sciences 2019, 224, 109-119; Babaei, et al., J Food Sci 2018, 83, (9), 2280-2287; de Andrade Teles, et al., Oxid Med Cell Longev 2018, 2018, 7043213) and recently reported studies using several AD models were summarized (Table 1)”, which includes PMID: 25666032.

Point 5-1: The molecular mechanism underlying the protective role of quercetin in AD is proposed via the antioxidant pathways by other reviews (PMID: 30914316, PMID: 26904161), which should be mentioned and discussed in this review.

Response 5-1: We are sorry for missing these citations. We cited the papers (PMID: 30914316, PMID: 26904161) in the revised text (p 1) and mentioned that “There is evidence suggesting that quercetin exhibits antioxidant and anti-inflammatory activities, as well as anti-tumor properties [9], and can improve several pathological conditions such as diabetes [10] and AD through regulation of anti-oxidative stress enzymes via the action of the nuclear factor (erythroid-derived 2)-like 2 (Nrf2) and antioxidant effect of paraoxonase 2 (PON2) expression [11, 12]”.

Point 5-2: This current review tries to link a novel mechanism pathway of quercetin via IRS inhibition, based on their previous finding that quercetin suppress ATF4 expression in an AD and diabetic mouse model (Ref. 52). Does quercetin suppress ATF4 in aged but non-diabetic AD mice?

Response 5-2: Yes. We mentioned in the revised text (p 7) that “This reduction in ATF4 expression was observed in the brain of APP23 mice fed quercetin for more than one year [47].----- the deterioration in memory was delayed in aged but non-diabetic AD fed with quercetin (aged 6-12 months). Contextual and auditory fear memories were enhanced in aged wild type mice fed with quercetin [47]”.

Point 5-3: other evidence that quercetin may regulate memory of AD or non-AD model through the IRS pathway should be stated with correlated reference.

Response 5-3: We mentioned in the revised text (p 8) that “In high-cholesterol-fed old mice, quercetin improves cognitive impairment, suppressing eIF2α phosphorylation [91]”.

Point 6: It is best to include a paragraph of “conclusion and Perspective” in the end of the review article.

Response 6: We agree with the Reviewer’s comment. We mentioned in the revised text of the Conclusion and Perspective (p 9) that “The ISR mediates cognitive impairments in mouse models of AD. The signaling molecules of the ISR were increased in post-mortem AD brains and other neurological diseases such as traumatic brain injury, amyotrophic lateral sclerosis, Huntington’s disease, Parkinson’s disease [115]. Inhibition of ISR signaling by quercetin rescues memory deficits in mouse models of AD and ISRIB prevents cognitive deficits in traumatic brain injury [116] and neurodegeneration in prion-disease mice [117]. However, identification of the target molecules involved in the ISR signaling in these neuronal diseases for treatment remains elusive. Future experiments should explore the main molecules for potential therapeutic interventions such as eIF2α kinases, eIF2B, and phosphatases (PP1c, CReP, GADD34). In addition, there are currently no markers that allow the monitoring of ISR signaling in the brain. Identification of potential markers is important for validation of the effect of quercetin in clinical trials because a pilot study showed that early stage AD patients improved their memory recall after the intake of onion powder containing high amounts of quercetin, for 4 weeks using the Revised Hasegawa Dementia Scale [114]. In the future, ISR signaling markers will be useful to investigate whether the intake of quercetin-rich onion powder by patients with mild cognitive impairments (MCI) modifies the progression of their symptoms.”

Point 7: The abstract stated that IRS signaling in human AD patients is also reviewed, but there is no clearly part about human study reviewed in the text. 

Response 7: We agree with the Reviewer’s comment. Since we mentioned in the revised text (p 4) that “Phosphorylated PERK is detected in the hippocampus and the temporal lobe of AD patients by immunohistochemistry [2]. Phosphorylation of eIF2α has also been detected in the brain of AD patients by immunohistochemistry [41] and western blot analysis [3]”, we have deleted “ and AD patients” in the abstract.

The following is a point-by-point change made to the in-text references and table.

Changes to text references

New references added in the revised text.

             [7]: Buffington, S. A., et al., Annu Rev Neurosci 2014, 37, 17-38.

             [8]: Oliveira, M. M., et al., , J Neurosci 2016, 36, (4), 1053-5.

             [11]: Zaplatic, E., et al., Life sciences 2019, 224, 109-119.

             [12]: Costa, L. G., et al., Oxid Med Cell Longev 2016, 2016, 2986796.

             [63]: Babaei, F., et al., J Food Sci 2018, 83, (9), 2280-2287.

[64]: de Andrade Teles, R. B., et al., Oxid Med Cell Longev 2018, 2018, 7043213.

             [69]: Sabogal-Guaqueta, A. M., et al., Neuropharmacology 2015, 93, 134-45.

             [91]: Lu, J., et al., The Journal of pathology 2010, 222, (2), 199-212.

[115]: Moon, S. L., et al., Trends Mol Med 2018, 24, (6), 575-589.

[117]: Halliday, M., et al., Cell death & disease 2015, 6, e1672.

Changes to table

Table 1 - the column references were amended and reference numbers in parentheses were added

Round  2

Reviewer 2 Report

The authors have done a good job responding to reviewer’s concerns and comments. The revised review has been quite improved, and structured better, more complete references have also been included.

Here are a few suggestions

Add the following sentence on page 1, line 42. After Ref. 13.

Recently, several studies suggest the effects of quercetin on memory and cognition improvement may be associated with ISR regulation (Ref.).

Page 6, Line 223-226. The beginning sentence is too abrupt.

Suggested change: Accumulating of evidence suggest the protective role for quercetin in cognitive decline and neurodegenerative disease (Ref. 11, 63, 64). Here, we summarize recently reported studies on the effects of quercetin using variety of AD models (Table 1), which showing quercetin can improve cognition and memory, and have beneficial effects on AD of different species.  

Conclusion and perspective

I do not quite agree with that the future direction is to identify potential ISR signaling marker, so that to investigate whether intake of quercerin-rich onion power by paptient to improve memory is through regulation of ISR. It is difficult to detect the ISR level in the brain of a patient (even you have a marker), and this is not of much therapeutic value. Therefore, this perspective is a kind of misleading, and need to be revised.

Author Response

Point 1: Add the following sentence on page 1, line 42. After Ref. 13. “Recently, several studies suggest the effects of quercetin on memory and cognition improvement may be associated with ISR regulation (Ref.).”

Response 1: We agree with the Reviewer’s comment. We added the sentence in the revised text (p 1) that “Recently, several studies suggest the effects of quercetin on memory and cognition improvement may be associated with ISR regulation [14-16].”.

Point 2: Page 6, Line 223-226. The beginning sentence is too abrupt. Suggested change: Accumulating of evidence suggest the protective role for quercetin in cognitive decline and neurodegenerative disease (Ref. 11, 63, 64). Here, we summarize recently reported studies on the effects of quercetin using variety of AD models (Table 1), which showing quercetin can improve cognition and memory, and have beneficial effects on AD of different species.

Response 2: We agree with the Reviewer’s comment and amended in the revised text (p 5) that “Accumulating of evidences suggest the protective role for quercetin in cognitive decline and neurodegenerative disease [11, 65, 66]. Here, we summarize recently reported studies on the effects of quercetin using variety of AD models (Table 1), which showing quercetin can improve cognition and memory, and have beneficial effects on AD of different species [67-73]. Karimipour et al.,---”.

Point 3: Conclusion and perspective. I do not quite agree with that the future direction is to identify potential ISR signaling marker, so that to investigate whether intake of quercerin-rich onion power by paptient to improve memory is through regulation of ISR. It is difficult to detect the ISR level in the brain of a patient (even you have a marker), and this is not of much therapeutic value. Therefore, this perspective is a kind of misleading, and need to be revised.

Response 3: We agree with the Reviewer’s comment. We deleted part of sentences “In addition, there are currently no markers that allow the monitoring of ISR signaling in the brain. Identification of potential markers is important for validation of the effect of quercetin in clinical trials because a pilot study showed that early stage AD patients improved their memory recall after the intake of onion powder containing high amounts of quercetin, for 4 weeks using the Revised Hasegawa Dementia Scale [114]. In the future, ISR signaling markers will be useful to investigate whether the intake of quercetin-rich onion powder by patients with mild cognitive impairments (MCI) modifies the progression of their symptoms.” in the revised text of the Conclusion and Perspective (p 9).

The following is a point-by-point change made to the in-text references and table.

Changes to text references

Number of references from page 2 to 9 amended in the revised text.

Changes to table

Table 1 - reference numbers amended

This manuscript is a resubmission of an earlier submission. The following is a list of the peer review reports and author responses from that submission.